# Purine Biosynthesis Pathways Are Required for Myogenesis in *Xenopus laevis*

**DOI:** 10.3390/cells12192379

**Published:** 2023-09-28

**Authors:** Maëlle Duperray, Fanny Hardet, Elodie Henriet, Christelle Saint-Marc, Eric Boué-Grabot, Bertrand Daignan-Fornier, Karine Massé, Benoît Pinson

**Affiliations:** 1Institut de Biochimie et Génétique Cellulaires, CNRS, UMR 5095, Université de Bordeaux, F-33000 Bordeaux, France; 2CNRS, IMN, UMR 5293, Université de Bordeaux, F-33000 Bordeaux, France

**Keywords:** metabolic disease, myogenic regulatory factors, hypaxial muscle progenitors, adenylosuccinate lyase, yeast model

## Abstract

Purines are required for fundamental biological processes and alterations in their metabolism lead to severe genetic diseases associated with developmental defects whose etiology remains unclear. Here, we studied the developmental requirements for purine metabolism using the amphibian *Xenopus laevis* as a vertebrate model. We provide the first functional characterization of purine pathway genes and show that these genes are mainly expressed in nervous and muscular embryonic tissues. Morphants were generated to decipher the functions of these genes, with a focus on the adenylosuccinate lyase (*ADSL*), which is an enzyme required for both salvage and de novo purine pathways. *adsl.L* knockdown led to a severe reduction in the expression of the myogenic regulatory factors (MRFs: Myod1, Myf5 and Myogenin), thus resulting in defects in somite formation and, at later stages, the development and/or migration of both craniofacial and hypaxial muscle progenitors. The reduced expressions of *hprt1.L* and *ppat*, which are two genes specific to the salvage and de novo pathways, respectively, resulted in similar alterations. In conclusion, our data show for the first time that de novo and recycling purine pathways are essential for myogenesis and highlight new mechanisms in the regulation of MRF gene expression.

## 1. Introduction

Purine triphosphate nucleotides, ATP and GTP are synthesized via a highly conserved [1] de novo pathway, allowing for sequential construction of the purine ring on a ribose phosphate moiety provided by phosphoribosyl pyrophosphate (PRPP) (Figure 1A and Appendix A for details). This de novo pathway results in the synthesis of inosine 5′-monophosphate (IMP), which can be converted into either AMP or GMP and then into other phosphorylated nucleotide forms required for all known forms of life. These nucleotides can alternatively be synthetized via a salvage pathway using precursors taken up from the extracellular medium or coming from the internal recycling of preexisting purines (Figure 1A). Purines and their derivatives (such as NAD(P) (nicotinamide adenine dinucleotide (phosphate)), FAD (Flavin adenine dinucleotide), coenzyme A, ADP-ribose, S-adenosyl-methionine and S-adenosyl-homocysteine) form a family of metabolites among the most abundant (mM range) in cells and are involved in a myriad of physiological events. Purines are required for numerous cellular processes, such as the biosynthesis of nucleic acids and lipids; replication; transcription; translation; maintenance of energy and redox balances; and regulation of gene expression (methylation, acetylation, etc.) and cell signaling, such as the purinergic signaling pathway [2]. Therefore, defects, even minor ones, in the purine de novo biosynthesis or recycling pathways lead to deleterious physiological effects. Indeed, to date, no less than 35 rare genetic pathologies have been associated with purine metabolism dysfunctions [3,4,5,6,7].

Purine-associated pathologies share a broad spectrum of clinical symptoms, including hyperuricemia, immunological, hematological and renal manifestations, as well as severe muscular and neurological dysfunctions [3,4]. They are all characterized by abnormal levels of purine nucleotides, nucleosides and/or nucleobases in the patient’s body fluids and/or cells. Although in most cases, the mutated gene has been identified, the causal link between the defective enzyme, the decrease in some final products, the accumulation of purine intermediates or the alteration of physiological functions with the observed symptoms often remains very elusive, even unknown. 

The adenylosuccinate lyase deficiency (OMIM 103050) is one of the most studied purine metabolism pathologies in which the functions of the Adsl enzyme, which catalyzes two non-consecutive reactions required both for the salvage and de novo purine pathways, are altered (Figure 1A). First described in 1984 [9], this rare autosomal recessive disorder is associated with a massive accumulation of succinyl derivatives (Succinyl-adenosine and Succinyl-AICAR (AminoImidazole CarboxAmide Ribonucleoside)) corresponding to dephosphorylation of the monophosphate substrates of Adsl (SZMP and SAMP, Figure 1A) in patient fluids (blood, urine and cerebrospinal fluid). The first patient-specific mutation in the *ADSL* gene was identified in 1992 [10] and, to date, more than 50 different mutations have been described [11] (Appendix A, orange and yellow boxes). All the biochemically studied mutations lead to a low residual Adsl enzymatic activity [12,13,14,15,16]. This pathology is characterized by serious neuromuscular dysfunctions, including psychomotor retardation, brain abnormalities, autistic features, seizures, ataxia, axial hypotonia, peripheral hypotonicity, muscular wasting and growth retardation [14,17,18]. Three distinct forms of *ADSL* deficiency have been categorized based on the severity of the symptoms: a fatal neonatal form (respiratory failure causing death within the first weeks of life), a childhood form with severe neuromuscular symptoms (severe form; type I) and a more progressive form with milder symptoms (mild form; type II) (for review see [19]). A robust correlation has been clearly established between the residual activity of Adsl, the consequent accumulation of succinyl derivatives and the intensity of phenotypes [16,20,21,22]. However, these studies do not explain the molecular bases associated with *ADSL* deficiency, though they raise a real interest in identifying biological functions that could be modulated by SZMP, SAMP and their derivatives. For example, SZMP acts as a signal metabolite that regulates transcriptional expression in yeast [8] and promotes proliferation in cancer cells by modulating pyruvate kinase activity [23], while SAMP stimulates insulin secretion in pancreatic cells [24]. More recently, RNA-seq analysis of a cellular model of *ADSL* deficiency identified the misexpression of genes involved in cancer and embryogenesis, bringing some first clues to understanding the molecular bases of this rare disease [25].

To go further into the etiology of this disease, it is now essential to identify the different biological processes that are altered by a purine deficiency during embryonic development, as the most severe symptoms appear in utero or within the first weeks or months after birth. To our knowledge, few invertebrate and vertebrate models have been developed to study this disease and no mammalian model recapitulating *ADSL* deficiency exists, most likely due to embryonic lethality [26,27]. Here, we established an *X. laevis* model to assess the developmental functions of *ADSL* by knocking down its embryonic expression. This powerful vertebrate model organism has been widely used to identify developmental impairments associated with human pathologies [28,29]. 

We report the identification and functional validation of the *adsl.L* gene and show this gene as being mostly expressed in the nervous and muscular tissues during *X. laevis* development. The knockdown of *adsl.L* results in downregulation at different developmental stages of several myogenic regulating factor (MRF) expressions, such as Myod1, Myf5 and Myogenin, and leads to alterations in somites, craniofacial and hypaxial muscle formation, thus showing that *adsl.L* is essential for myogenesis. Finally, functional comparative analysis of other enzymes in the de novo and salvage purine pathways (*ppat.L*, *ppat.S* and *hprt1.L*) highlights a major role for purine metabolism in muscle tissue development, providing insight into the developmental defects that underlie some of the symptoms of patients with severe ADSL deficiency, as well as other purine-associated pathologies.

## 2. Materials and Methods

### 2.1. Yeast Media 

SDcasaW is an SD medium (0.5% ammonium sulfate, 0.17% yeast nitrogen base without amino acids and ammonium sulfate (BD-Difco; Franklin Lakes, NJ, USA) and 2% glucose) supplemented with 0.2% casamino acids (#A1404HA; Biokar/Solabia group; Pantin, France) and tryptophan (0.2 mM). When indicated, adenine (0.3 mM) or hypoxanthine (0.3 mM) was added as an external purine source in SDcasaW. 

### 2.2. Yeast Strains and Plasmids

All yeast strains are listed in Appendix A and belong to or are derived from a set of disrupted strains isogenic to BY4741 or BY4742 purchased from Euroscarf (Oberursel, Germany). Double mutant strains were obtained via crossing, sporulation and micromanipulation of meiosis progeny. All plasmids (Appendix A) were constructed using the pCM189 vector [30], allowing for the expression of the *X. laevis* gene in yeast under the control of a tetracycline-repressible promoter. *X. laevis* open reading frame (ORF) sequences were amplified using PCR from I.M.A.G.E Clones (Source Biosciences; Nottingham, UK). All *X. laevis* ORF sequences were fully verified via sequencing after cloning in the pCM189 vector. Further cloning details are available upon request. 

### 2.3. Yeast Growth Test 

For the drop tests, yeast transformants were pre-cultured overnight on a solid SDcasaWA medium, re-suspended in sterile water at 1 × 10^7^ cells/mL and submitted to 1/10 serial dilutions. Drops (5 μL) of each dilution were spotted on freshly prepared SDcasaW medium plates supplemented or not with adenine or hypoxanthine. Plates were incubated either at 30 or 37 °C for 2 to 7 days before imaging.

### 2.4. Embryo Culture

*X. laevis* males and females were purchased from the CNRS Xenopus Breeding Center (CRB, Rennes, France). Embryos were obtained via in vitro fertilization of oocytes collected in 1× Marc’s Modified Ringers saline solution (1× MMR: 100 mM NaCl, 2 mM KCl, 2 mM CaCl_2_, 1 mM MgSO_4_, 5 mM Hepes, pH 7.4) from a hormonally (hCG (Centravet, Nancy, France), 450 units) stimulated female by adding crushed testis isolated from a sacrificed male. Fertilized eggs were de-jellied in 3% L-cysteine hydrochloride, pH 7.6 (Sigma-Aldrich; Merck group, Darmstadt, Germany), and washed several times with 0.1 × MMR. Embryos were then cultured to the required stage in 0.1× MMR in the presence of 50 µM of gentamycin sulfate. Embryos were staged according to the Nieuwkoop and Faber table of *X. laevis* development [31].

### 2.5. mRNA Synthesis and Morpholino Oligonucleotides

Capped mRNAs were synthesized using Sp6 mMESSAGE mMACHINE Kits (Ambion, Thermo Scientific, Waltham, MA, USA) from linearized plasmids (listed in Appendix A). *adsl.L*, *ppat.S*, *ppat.L* and *hprt1.L* RNA were transcribed from the IMAGE clones. *adsl.L*-RNA*, *ppat.S*-RNA*, *ppat.L*-RNA* and *hprt1.L** ORFs were amplified using PCR in conditions that allow for the introduction of mutations in the morpholino oligonucleotide (MO) binding site and were subcloned into the plasmid pBF. *Homo sapiens ADSL cDNA* was subcloned into the plasmid *pCS2+. adsl.L* MO1 (5′-AAGCATGGAGGGGAGCAGTGGGCTAAG-3′), *adsl.L* MO2 (5′-ATGGAGGGGAGCAGTGGGCTAAGCAT-3′), *hprt1.L* MO (5′-GGACACAGGCTCAGACATGGCGAGC-3′), *ppat.L*/*ppat.S* MO (5′-GTGATGGAGTTTGAGGAGCTGGGGAT-3′) and standard control MO (cMO) were designed and supplied by Gene Tools, LLC. The position of the MOs in relation to their respective RNA is indicated in Appendix A. 

### 2.6. Microinjections

Embryos were injected with MO alone or in combination with MO non-targeted mRNA (mutated *adsl.L* RNA* or *Homo sapiens ADSL*, as specified in the text/legend) into the marginal zone of one blastomere at the 2-cell stage. *LacZ* (250 pg) RNA was co-injected as a lineage tracer. Embryos were injected in 5% Ficoll, 0.375× MMR, cultured to various developmental stages, fixed in MEMFA (MOPS 100 mM pH 7.4, EGTA 2 mM, MgSO_4_ 1 mM, 4.0% (*v*/*v*) formaldehyde) and stained for β-galactosidase activity using Red-Gal or X-Gal substrates (RES1364C-A103X or B4252, Merck) to identify the injected side and correctly targeted embryos. Embryos were fixed again in MEMFA before dehydration into 100% methanol or ethanol for immunohistochemistry or in situ hybridization, respectively. 

### 2.7. In Situ Hybridization

Whole-mount in situ hybridizations (ISHs) were carried out as previously described [32,33,34]. Sense and antisense riboprobes were designed by subcloning fragments of coding cDNA sequences in pBlueScript II (SK or KS) plasmids (Addgene, Watertown, MA, USA). Riboprobes were generated by in vitro transcription using the SP6/T7 DIG RNA labeling kit (Roche, Basel, Switzerland, #11175025910) after plasmid linearization, as indicated in Appendix A. Riboprobe hybridization detection was carried out with an anti-DIG Alkaline Phosphatase antibody (Roche, #11093274910) and the BM-Purple AP substrate (Roche, #11442074001). Riboprobes for *lbx1*, *mespa*, *myf5*, *myod1*, *myogenin*, *pax3*, *tbxt* (*xbra*) and *tcf15* were previously described [35,36,37]. 

### 2.8. Immunostaining 

Whole-mount immunostaining of differentiated skeletal muscle cells was performed using the monoclonal hybridoma 12-101 primary antibody [38] (1/200 dilution; DSHB #AB-531892) and the EnVision^+^ Mouse HRP kit (Agilent Technologies, Santa Clara, CA, USA, K4007) according to the manufacturer’s recommendations. 

### 2.9. Temporal Expression of Genes Established Using RT-PCR

RNA extraction from whole embryos, cDNA synthesis and RT-PCR were performed as previously described [33,39]. Sequences of the specific primers designed for each gene and PCR amplification conditions are given in Appendix A. Chosen primers were selected to differentiate homeolog gene expression and to discriminate potential genomic amplification from cDNA amplification. PCR products were verified via sequencing (Eurofins genomics). The quantity of input cDNA was determined via normalization of the samples with the constant ornithine decarboxylase gene *odc1.L* [40]. Linearity of the signal was controlled by carrying out PCR reactions on doubling dilutions of cDNA and negative controls without either RNAs, reverse transcriptase or cDNA were also performed. Experiments were done at least twice on embryos from two different females (N ≥ 2) and representative profiles are shown in Figure 2. 

### 2.10. Embryo Scoring and Photography

Embryos were bleached (1% H_2_O_2,_ 5% formamide, 0.5× SSC) to remove all visible pigment, and phenotypes were determined in a commonly used way, i.e., blind-coded, by comparing the injected and un-injected sides. Only embryos with normal muscle tissue formation on the un-injected side and correctly targeted β-galactosidase staining on the injected side were scored. Transverse sections were performed with a razor blade on fixed embryos. Embryos were photographed using an SMZ18 binocular system (Nikon, Minato City, Tokyo, Japan).

### 2.11. Statistics and Reproducibility 

All experiments were carried out on at least two batches of fertilized eggs from two independent females (N ≥ 2). Histograms represent the percentage of embryos displaying each phenotype and the number of embryos in each category is indicated in the bars of each histogram. Fisher’s exact test was used for the statistical analyses and *p*-values are presented above the bars of the histogram in all figures. 

### 2.12. Bioinformatics

Sequences were identified on the NCBI and Xenbase databases [41]. Basic Local Alignment Search Tool (BLAST) searches were performed on the NBCI Nucleotide and the Xenbase *X. laevis* 9.1 Scaffolds genome databases [42]. Conceptual translation of complementary DNA (cDNA) was performed on the ExPASy Internet website using the program Translate Tool (web.expasy.org/translate/, accessed on 1 September 2017). A comparison of the coding sequences from *H. sapiens*, *S. cerevisiae*, *X. laevis* and *X. tropicalis* was done on https://blast.ncbi.nlm.nih.gov (2022). Accession numbers of all sequences used in this study are given in Appendix A.

## 3. Results

### 3.1. Identification of X. laevis Purine Pathway Genes 

The purine biosynthesis pathways are known to be highly conserved throughout evolution [1]. Most of *X. laevis* genes encoding potential purine-pathway enzymes have been putatively identified and annotated via automated computational analyses from the entire genome sequencing [43]. Protein sequences were deduced from the conceptual translation of these annotated genes and aligned with their potential orthologs, e.g., *H. sapiens* and *X. tropicalis* (Appendix A). The vast majority of *X. laevis* de novo and salvage pathways have two homeologs, whose protein sequences display a high degree of identity with their *X. tropicalis* and human orthologs (more than 85 and 50%, respectively; Figure 1A and Appendix A), suggesting that these annotated genes effectively encode the predicted purine pathway enzymes. 

To establish that the putative *X. laevis* purine pathway genes encode the predicted enzymatic activities, a functional complementation assay was undertaken via heterologous expression of *X. laevis* genes in the yeast *Saccharomyces cerevisiae,* which was shown to be a valuable model to investigate metabolic pathways [44]. Yeast knockout mutants were available in the laboratory and sequence alignments showed a high degree of identity between *S. cerevisiae* and *X. laevis* orthologous protein sequences (Appendix A). Plasmids allowing for expression of the *X. laevis* ORF in *S. cerevisiae* were transformed in the cognate yeast knockout mutant, and functional complementation was tested. As shown in Figure 1B, the yeast *adsl* knockout mutant (*ade13*) was unable to grow in the presence of hypoxanthine as a unique purine source, but growth was restored by the expression of either the *S. cerevisiae ADE13* or the *X. laevis adsl.L* gene. By contrast, all these strains were able to grow in the presence of adenine, which allowed for purine synthesis via the adenine phosphoribosyl transferase (Apt1) and AMP deaminase (Amd1) activities (Appendix A). Similar experiments were conducted with 15 other *X. laevis* purine-pathway-encoding genes, 13 of which were able to complement the *S. cerevisiae* cognate knockout mutants (Appendix A). Altogether, this data allowed us to functionally validate *adsl.L* and 13 other *X. laevis* genes involved in purine de novo and recycling pathways, demonstrating that the purine pathways were functionally conserved in *X. laevis*. 

We then focused on the role of adenylosuccinate lyase, as (1) it is encoded by a single gene (*adsl.L)*, greatly simplifying the knockdown experiments, and its protein sequence is highly conserved between human and *X. laevis* (Appendix A); (2) it is the only enzymatic activity that is required in both the de novo biosynthesis and recycling purine pathways; and (3) its mutation leads to severe developmental alterations, for which the molecular bases associated with the symptoms are still largely unknown.

### 3.2. The adsl.L Gene Was Mostly Expressed in Muscular and Neuronal Tissues and Their Precursors during X. laevis Development

*adsl.L* spatiotemporal expression during *X. laevis* development was established via two complementary approaches. First, its temporal expression profile was determined using RT-PCR on whole embryos from fertilized egg to stage 45 (Figure 2A). The *adsl.L* gene displays both a maternal (before the mid-blastula transition (MBT)) and a zygotic expression at all developmental stages studied. 

*adsl.L* spatial expression was then determined via in situ hybridization (Figure 2B). *adsl.L* expression is detected in the animal, but not vegetal, pole during the cleavage phase. Zygotic expression was detected in the neuroectoderm and the paraxial mesoderm during neurulation. From stage 25 until the late organogenesis stages, *adsl.L* expression was found in the developing epibranchial placodes, lens and in the central nervous system (fore–midbrain and mid–hindbrain boundaries). From stage 30 onward, *adsl.L* transcripts were also detected in the somites and its somitic expression appeared to increase during organogenesis. At the late organogenesis stage, its expression was detected in other mesoderm-derived tissues, such as the pronephric tubules (proximal and intermediate tubules) and the heart. 

Using similar approaches, we showed that (1) 14 out of the 16 tested genes display a similar temporal expression profile to that of *adsl.L* (Appendix A) and (2) all 9 genes tested using in situ hybridization were expressed in neuromuscular tissues (e.g., somites, hypaxial muscles, central nervous system and retinas), even if differences in the tissue specificity existed for some of these genes (Appendix A). These spatiotemporal expression patterns are consistent with the fact that neuromuscular dysfunctions are the primary symptoms associated with purine-dependent diseases. 

### 3.3. The adsl.L Gene Was Required for X. laevis Myogenesis 

To understand the role of *adsl.L* during muscle development, we undertook a loss of function approach using two specific anti-sense morpholino oligonucleotides (MOs) (see Appendix A for MOs efficiency). MO injections were performed in the marginal zone of one blastomere at the two-cell stage. This injection has the advantage of unilaterally affecting the development of the future mesoderm, with the non-injected side serving as an internal control. A dose-dependent curvature along the anteroposterior axis was observed following the injection of *adsl.L* MO1 and MO2, and not of the control morpholino (cMO), with a concave deformation always corresponding to the side of the injection (Figure 3A,B). This curvature phenotype was already documented when myogenesis was altered [45], suggesting that *adsl.L* gene could be required for proper muscle formation in *X. laevis*. To test this hypothesis, we performed immunostaining on tailbud and tadpole stages with the differentiated skeletal-muscle-cell-specific 12-101 antibody [38]. The cMO-injected embryos displayed normal global morphology and normal muscle phenotype at all stages tested. By contrast, myotomes were smaller along the dorsoventral axis following the injection of 10 ng of *adsl.L* MO1 or MO2 at the tailbud stage (Figure 3C,D). Furthermore, the somite morphology was considerably altered. Straight-shape somites, blurred somite domains and fewer v-shaped somites were observed on the injected side. The 12-101-positive area was also reduced along the anteroposterior axis, with the latter shortening most likely being at the origin of the curvature phenotype observed (Figure 3A). This muscle alteration was even more exacerbated by increasing the dose of MO, with severe reduction and, in some cases, absence of a 12-101-positive domain and no somitic-like structure observed on the injected side in more than 80% of the injected 20 ng MO1 embryos (Figure 3C). This dose-dependent muscle phenotype was still observed in the tadpole stage (stage 39), with a similar percentage of *adsl.L* morphant embryos displaying myotome reduction and altered chevron-shaped somites in their injected side, especially in the anterior trunk region (Appendix A). At the late tadpole stage (stage 41), somite defects, which were characterized by anterior straight or criss-crossed somites, were observed on the injected side of *adsl.L* morphants (red arrowheads; Figure 3E,F), whereas the cMO embryo injected side displayed the characteristic v-shaped somites (white arrowheads; Figure 3E,F). Reduction in the hypaxial muscles (precursors of the limb muscles) was also observed in *adsl.L* morphants but not in cMO-injected embryos (Figure 3E,G; compare the blue to the white dashed areas in *adsl.L* MO1 embryo). This hypaxial muscle phenotype was worsened when the MO dose was increased with no hypaxial muscle progenitors in 70% and 96% of analyzed embryos following 10 ng and 20 ng of *adsl.L* MO1 injection, respectively (Appendix A). These phenotypes were rescued by co-injecting *adsl.L* MO1 with the *adsl.L* mutated RNA*, whose translation was unaffected (Appendix A) by *adsl.L* MO1 and MO2 (Figure 3E–G). No significant phenotype was obtained by co-injecting the *adsl.L* RNA* with cMO (Figure 3E–G). Together, these results validate the muscular-associated phenotypes as a consequence of a specific *adsl.L* gene knockdown and not of the potential morpholino injection’s side effects. 

To confirm the *adsl.L* gene implications during myogenesis, the expressions of *tcf15* (*paraxis*) and *mespa* (*thylacine* 1) genes were analyzed in *adsl.L* MO1 and MO2 morphants at late neurula stages (stages 17–19). As expected, *tcf15* expression was observed in the dorsolateral part of the somites [35] following the control cMO injection. However, its expression domain was reduced following *adsl.L* knockdown (Figure 4A,B), suggesting that the primitive myotome formation was impacted in *adsl.L* morphants. Alterations in the expression of *mespa*, which is a somitomere marker [46,47], were also observed in *adsl.L* knockdown conditions (Figure 4C,D). These alterations included a reduction in, and even an absence of, mespa stripes number and their lateral length. The most frequent phenotype was a shift to a more anterior position in the presomitic mesoderm (white arrows, Figure 4C). These significant phenotypes suggest that *adsl.L* was involved in the segmentation of the presomitic mesoderm. 

To determine whether the observed muscle phenotypes were specifically related to the knockdown of the *adsl.L* gene or to the purine pathways alteration, similar loss of function experiments (see Appendix A for the MOs efficiencies) were carried out on other purine-associated genes: (1) the *ppat* (*ppat.L* and *ppat.S*) genes encoding the phosphoribosylpyrophosphate amidotransferase (Ppat), which is the first enzyme of the purine de novo pathway, and (2) the *hprt1.L* gene encodin Hypoxanthine-PhosphoRybosylTransferase (Hprt) in the salvage pathway (Figure 1A), which is the enzyme that allows for metabolization of hypoxanthine, which is the major purine source in *X. laevis* embryos [48]. Comparable muscle alterations were obtained with independent knockdown of these genes, i.e., a strong alteration of somite shape at both the tailbud and tadpole stages, which were associated with smaller myotomes at the early organogenesis stage and reduction in, or even absence of, hypaxial muscles in the latter stages (Appendix A). Myogenesis alterations were also observed at the neurula stages (Appendix A). A reduction in the *tcf15* expression domain was observed following *ppat* and *hprt1.L* MO injection (Appendix A). Furthermore, the formation of somitomeres was disrupted in *ppat* and *hprt1.L* morphants, which showed a shift and number reduction of mespa stripes on the injected side (Appendix A).

Taken together, these similar muscle phenotypes observed in *adsl.L*, *ppat* and *hrpt1.L* morphants demonstrated a strong association between altered purine pathways and defects in somitogenesis and myogenesis.

### 3.4. Expression of the myod1 and myf5 Genes Required Functional Purine Pathways 

We assessed whether the muscular phenotypes observed upon the knockdown of purine-associated genes could result from an alteration of the myogenic regulatory factors’ (MRFs’) expression during the different myogenic waves [36,49]. The expression of both *myod1* and *myf5* was first analyzed via in situ hybridization at the early neurula stage, during the first myogenic wave (Figure 5A–D). A strong reduction in both *myod1* and *myf5* expression was observed on the injected side of *adsl.L* morphants (Figure 5A,C), whereas no alteration could be seen in the cMO-injected embryos. This significant reduction was even stronger in the anterior region, which gave rise to the first somites. Importantly, *myod1* expression was rescued by the injection of *X. laevis adsl.L* mutated RNA* or human *ADSL* RNA in *adsl.L* MO1 and MO2 morphants (Figure 5A,B). These results are consistent with an adenylosuccinate lyase activity, even when heterologous, which is required for the proper expression of *myod1*. Of note, the overexpression of either *X. laevis adsl.L* mutated RNA* or human *ADSL* mRNA induced a small but significant increase in *myod1* expression (Appendix A). To exclude the idea that the observed reduction of *myod1* and *myf5* expression resulted from an alteration of mesoderm formation and/or integrity in *adsl.L* morphants, expression of the pan-mesoderm marker *tbxt* (*xbra*) was analyzed via in situ hybridization on whole embryos at stage 11. No significant difference was observed in all injected embryos (Figure 5E). Finally, a strong and similar reduction in *myod1* expression, with no alteration of the *tbxt* gene expression pattern, was also observed in either *ppat.L/ppat.S* or *hprt1.L* morphants (Appendix A). Altogether, these results show that functional purine synthesis is strictly required for a proper expression of the myogenic regulatory factor genes *myod1* and *myf5* during the first myogenic wave.

To further analyze the effects of *adsl.L* knockdown on MRF expression, *myod1* and *myf5* expression was then analyzed at later developmental stages (Figure 6). At the tailbud stage, the *myod1* expression in the somites was found to be markedly altered in the *adsl.L* MO1 and MO2 morphants. As observed with the 12-101 staining, myotomes were shortened along the dorsoventral axis and somite shape defects and even a loss of the chevron v-shape of the most anterior somites were observed (blue arrowheads, Figure 6A,B). *myod1* expression in the ventral border of the dermomyotome was also markedly reduced or lost (red arrowheads, Figure 6A,B), while its expression in the dermomyotome dorsal border was less altered in the anterior trunk region. The expression of *myod1* in craniofacial muscles was also reduced or even absent following *adsl.L* MO1 or MO2 injection (green arrowheads, Figure 6A,B). These alterations in craniofacial muscles were also observed at the tadpole stages, with a severe significant reduction in *myod1* expression, especially in the interhyoideus (ih), quadrato-hyoangularis and orbitohyoideus anlagen (q/oh) in *adsl.L* MO1 and MO2 morphants (light green arrowheads, Figure 6C,D). The expression of *myf5* was also noticeably altered in craniofacial muscles at both the tailbud and tadpole stages (Figure 6E–H), even if these alterations were milder than the *myod1* ones. Indeed, *myf5* expression in *adsl.L* morphants was reduced or absent at both stages in the dorsal region of the pharyngeal arch muscle anlagen (black arrowhead, Figure 6G), whereas its expression in the interhyoideus and the intermandibularis muscle anlagen was rarely reduced (pink arrowhead, Figure 6G). However, no alteration in *myf5* expression was observed in the undifferentiated presomitic mesoderm in the tail region (yellow circles, Figure 6E–H). The implication of *adsl.L* in the formation of the cranial muscles was also confirmed by analyzing the expression of myogenin (*myog*) following *adsl.L* MO1 and MO2 injection at stage 39 (Figure 7A and Appendix A). A strong reduction or even an absence of *myogenin* staining in the pharyngeal arch muscle anlagen was observed on the injected side in *adsl.L* morphants, whereas its expression in the interhyoideus and intermandibularis muscle anlagen was, however, less affected. These results are consistent with those observed via 12-101 immunolabelling on *adsl.L* morphants (Appendix A, blue arrowhead). Altogether, these data show that *adsl.L* is required for *myod1*, *myf5* and *myogenin* expression during the second wave of myogenesis. 

### 3.5. The adsl.L Gene Was Required for Hypaxial Muscle Formation and Migration 

In addition to the somite and craniofacial muscle phenotypes, severe hypaxial muscle alterations were observed in *adsl.L* morphants (Figure 3 and Appendix A). To further characterize these defects, the expression of *myogenin* was analyzed via in situ hybridization at the late tadpole stage. *adsl.L* knockdown resulted in either a strong decrease or even an absence of *myog* expression in hypaxial muscle cells, while no significant alteration was observed in the embryos injected with the control morpholino (cMO) in combination or not with the *adsl.L* mutated RNA* (Figure 7A,B). This muscle phenotype was rescued by the injection of the *adsl.L* mutated RNA* in *adsl.L* MO2 morphants. These *myogenin* expression alterations were confirmed, as the *myod1* expression was strongly reduced or absent in hypaxial muscles in *adsl.L* morphants (purple arrowheads, Figure 6C and Appendix A). These results confirm the role of the *adsl.L* gene in the expression of myogenic regulatory factors required for correct hypaxial muscle development. In addition, hypaxial muscles were also found to be highly reduced or absent, as revealed via 12-101 immunolabelling in *ppat.L/ppat.S* and *hprt1.L* morphants (pink arrowheads, Appendix A). Interestingly, hypaxial muscle defects were also observed in *tcf15* knockout mice [50], which is a gene whose expression is altered in purine *X. laevis* morphants (Figure 4A).

To better understand the deleterious effect of *adsl.L* knockdown on hypaxial muscles, expression of the *pax3* and *lbx1* genes encoding transcription factors were analyzed at different tadpole stages. *Lbx1* and *pax3* were both expressed in hypaxial body wall undifferentiated cells that form the front of hypaxial myoblast migration [37,51]. Although the injection of *adsl.L* MOs, especially *adsl.L* MO2, induced a slight reduction in *lbx1* expression in the hypaxial muscle precursors, the major observed phenotype was a migration alteration of these *lbx1* positive cells (Figure 7C,D). Indeed, muscle progenitors were closer to the ventral border of the somites. This alteration was more pronounced in the most anterior region (trunk I), in which progenitors migrated around the hyoid region (see yellow double arrows in Figure 7C). Similar alterations were observed for *pax3* positive hypaxial body wall progenitors, which were characterized by a severe alteration of hypaxial progenitor migration, while the *pax3* expression was less or not affected in the hypaxial myotome (Figure 7E,F). Altogether, these results show that purine biosynthesis genes are essential during hypaxial muscle formation in *X. laevis*. 

## 4. Discussion

### 4.1. X. laevis as a Vertebrate Model to Study Purine-Deficiency-Related Pathologies

In this study, we functionally characterized the main members of the purine biosynthesis and salvage pathways in the vertebrate *X. laevis* for the first time. We also established the comparative map of their embryonic spatiotemporal expression and showed that the main common expression of purine genes was in the neuromuscular tissue and its precursors. This expression profile could be directly linked to the neuromuscular symptoms usually observed in most of the nearly thirty identified purine-associated pathologies [3,4,5,6,7]. Furthermore, we investigated the embryonic roles of three purine biosynthesis enzymes during vertebrate muscle development, focusing on the Adsl.L enzyme. As *ADSL* deficiency in humans is associated with residual enzymatic activity (reviewed in [6,52]), we believe that our experimental strategy based on the knockdown with morpholino-oligonucleotides is more appropriate in identifying the molecular dysfunctions found in patients rather than strategies based on knockout experiments. Altogether, our data show that *X. laevis* is a relevant model for studying the molecular bases of the developmental alterations associated with purine deficiencies.

### 4.2. Role of Purine Pathway Genes during X. laevis Development

Maternal and zygotic expression are key indicators of potential pathway involvement during vertebrate development. We show that the early stages of *X. laevis* embryo development are dependent on both purine pathways, as all the genes studied, except for *adss1.L* and *adss1.S,* are maternal genes and are expressed from gastrula stages. However, from the early organogenesis phase (stage 22), all 17 tested genes were expressed. We previously showed that the unique reserve of purine in embryos, namely, hypoxanthine, is depleted at stage 22 [48], and thus, this embryonic stage is placed as a key turning point of the purine source for the embryo. From this stage onward and until stage 45 (the stage where embryos start to feed), purines can only be synthesized de novo or recycled within the embryo, and the zygotic expression of purine biosynthesis genes then becomes crucial. 

Furthermore, the genes of the purine de novo and salvage pathways share simultaneous embryonic expression and mostly in the same tissues, strongly suggesting that neither pathway is preferred for purine supply during *X. laevis* development. Our functional experiments also demonstrated that both pathways were involved in development since the muscle alterations observed in *adsl.L* morphants were also observed in *ppat.L/ppat.S* and *hprt1.L* morphants. This is different from what was obtained in *C. elegans*, in which muscle phenotypes associated with *ADSL* deficiency are rather dependent on the purine recycling pathway [27]. 

In addition, we showed that *adsl.L* plays an early and essential role in myogenesis during the first and second myogenic wave, and its knockdown is associated with strong alterations in the primary myotome, somitomeres and anterior somite formation and the subsequent malformation of hypaxial and craniofacial muscles. Embryonic neuromuscular defects were previously described in other animal models [26,27], with a strong alteration in muscle integrity observed in *C. elegans adsl* knockout mutants. These studies raised the question of whether the observed phenotypes were the result of a decrease in one or more end products of the purine biosynthetic pathways, or that of an accumulation of Adsl.L substrates or their derivatives. Indeed, the neurodevelopmental alterations observed in zebrafish *adsl* mutants were associated with the accumulation of SAICAr, which is the dephosphorylated form of Adsl substrate in the de novo pathway (SAICAR or SZMP), and thus, these neuronal alterations were rescued via inhibition of this pathway [26]. In *X. laevis*, as the alterations in myogenesis are very similar in the *adsl.L*, *ppat.L*/*ppat.S* and *hprt1.L* morphants, it seems very unlikely that the muscle phenotypes were related to an accumulation of Adsl.L substrates or their derivatives (Figure 1A). A metabolic rescue of the phenotypes is, therefore, an appropriate strategy to determine whether the muscle phenotypes could be related to a decrease in Adsl.L products and/or downstream metabolites. Hypoxanthine, which is the main source of purine in *X. laevis* embryo [48], would have been the best candidate metabolite for the MO rescue, but it could not be tested here since its metabolization requires the enzymatic activity of Adsl.L (Figure 1A). Rescue experiments were therefore carried out by adding adenine to the *adsl.L* morphant culture medium in concentrations at which this purine precursor was soluble in the medium (50 µM) but without any success. We cannot rule out that this may be due to the incapacity of this purine to cross the vitelline membrane. However, rescue experiments via adenine injection into the blastocoel or archenteron, as previously performed for other purine [32,53], are not possible because of adenine’s low solubility in aqueous solutions (<few µM).

Overall, these results strongly suggest that the observed muscle phenotypes are associated with purine deficiency, regardless of the mode of biosynthesis of these purines since disruption of the de novo pathway (*ppat.L*/*ppat.S* MO), the recycling pathway (*hprt1.L* MO) or both (*adsl.L* MO) results in similar alterations in myogenesis. However, these alterations may explain the muscle symptoms, such as axial hypotonia, peripheral hypotonia and muscle wasting observed in patients [14,17,18]. 

### 4.3. Muscle Defects Associated with Purine Pathways Dysfunction Were Related to Altered Expressions of MRF Genes 

Our study provides new in vivo evidence for the identification of altered molecular mechanisms during purine deficiency. Indeed, we show that functional purine pathways were required for the expression of the myogenic regulator factors Myod1, Myf5 and Myogenin, which are by far the master regulatory transcription factors of the embryonic myogenic program, including myotome and somite formation and hypaxial and craniofacial myogenesis [36,49]. The *adsl.L*-specific alteration of *myod1* and *myf5* expression in the ventral border of the dermomyotome could, therefore, be at the origin of the deleterious effects observed in morphants on hypaxial progenitors and differentiated hypaxial muscle cells. A strong decrease in MRF expression was observed in the *adsl.L* morphants from the earliest stages of muscle formation in the neurula embryos. As MRF genes regulate the expression of other MRFs and/or their own expression, it is possible that one (or more) key purine derivative(s) could regulate the transcription expression and/or the function of one of the MRFs. Interestingly, the opposite phenotypes were obtained for *myod1* expression in *adsl.L* morphants and in embryos overexpressing this gene, placing *adsl.L* as a key regulator of *myod1* expression during development. Since Myod1 is a potent inducer of muscle differentiation and is required for *myf5* expression in early *X. laevis* development, correct anterior somite segmentation [54] and correct myotome size [35], it is, therefore, possible that the muscle defects in *adsl.L* morphants were mainly due to a deregulation of *myod1* expression, thus placing this gene, and the purine biosynthetic pathways, at the top of the myogenic gene cascade. An interesting future point of investigation would be to test whether *myod1* expression is able to rescue *adsl.L* morphant muscle phenotypes.

So far, we have not found a direct or indirect link between purines and the transcriptional regulation of MRFs. However, regulation of transcription factor activity via direct binding of purine metabolites has already been demonstrated in different organisms [8,55]. In yeast, we have shown that the direct binding of purine metabolites (SZMP and ZMP, Figure 1) on Pho4 and Pho2 transcription factors was responsible for their activation [8]. These two metabolites of the de novo purine pathway were certainly not those implicated here, as similar muscle phenotypes were obtained in *adsl.L*, *ppat.L*/*ppat.S* and *hprt1.L* morphants, but we can speculate that another purine derivative could similarly be involved in modulating the transcriptional expression of MRFs during *X. laevis* myogenesis. Moreover, regulation of *myogenin* gene expression by metabolites (CMP and UMP) was recently observed in the mouse C2C12 cell line [56], showing that MRF expression can be modulated by nucleotides. To our knowledge, a similar direct regulation of MRFs by purines has not been described to date. If it exists, as suggested by our data, then the mechanism by which it acts remains to be established.

It may be possible that the effect of purines on MRF expressions is indirect via upstream cascades, responding to purines and regulating the expression of these genes. Indeed, many upstream signals, for example, Wnt and FGF, regulate the expression of MRFs in myogenesis (reviewed in [57]). These pathways involve the activation of protein kinases of the myogenic kinome, whose activity is highly dependent on intracellular pools of purine nucleotides [58]. In particular, the activity of protein kinase A, which is initiated by the activation of heteromeric G proteins and adenylate cyclase through the Wnt pathway, is required for the expression of *myod1* and *myf5* and the formation of myoblasts. In addition, extracellular triphosphate purine nucleotides are ligands of purinergic receptors that are involved in myoblast proliferation and differentiation [59,60]. We recently showed that the ATP-dependent P2X5 receptor subunit is specifically expressed in *X. laevis* somites, which is in agreement with its expression profile described in other animal models [61,62]. We may hypothesize that *adsl.L* knockdown may reduce the availability of extracellular purines, leading to a possible alteration in P2X5 activation and myogenesis impairment [62]. This hypothesis will be tested in the near future in the laboratory. We previously showed that purinergic signaling controls vertebrate eye development by acting on the expression of the PSED (*pax*/*six*/*eya*/*dach*) network genes [29]. As this PSED gene network regulates MRF expression [63], it may also be possible that a similar mechanism is involved in *adsl.L* morphants. Interestingly, our observed muscle phenotypes are highly similar to those induced by retinoic acid receptor β2 (RARβ2) loss of function, i.e., a rostral shift of presomitic markers, defects in somite morphology, chevron-shaped somites and hypaxial muscle formation [64]. Retinoic acid pathway, through the activation of RAR, regulates myogenic differentiation and MRF gene expression [65]. It was reported that retinoic acid affects the synthesis of PRPP in human erythrocytes in psoriasis [66] and RA deficiency interferes with the expression of genes involved in the purine metabolism pathway in vivo [67]. These data, along with ours, suggest a possible link between purine biosynthetic pathway, retinoic acid signaling pathway and MRF gene expression, which is worthy of further investigation in the future.

In conclusion, the establishment of this new animal model allowed us to demonstrate the critical functions of the purine biosynthetic pathways during vertebrate embryogenesis. Indeed, our results provide evidence for the roles of purine pathway genes in myogenesis through the regulation of MRF gene expression during embryogenesis. Although the first median and lateral myogenesis disappeared during evolution, the second myogenic wave is conserved [49]. We can, therefore, speculate that the involvement of the de novo and salvage purine pathways during this myogenic wave is conserved in mammals and that purines may, therefore, be key regulators in the formation of hypaxial myogenic cells, i.e., the progenitors of the abdominal, spinal, and limb muscles, those muscles which are affected in patients with purine-associated pathologies [4,5,7,52]. Although in this work, we mostly focused on muscle alterations, it provides the basis for further functional studies to identify the molecular mechanisms involved in the development of neuromuscular tissues, those tissues in which the alterations underlie the major deleterious symptoms of patients with purine deficiency.

## Figures and Tables

**Figure 1 cells-12-02379-f001:**
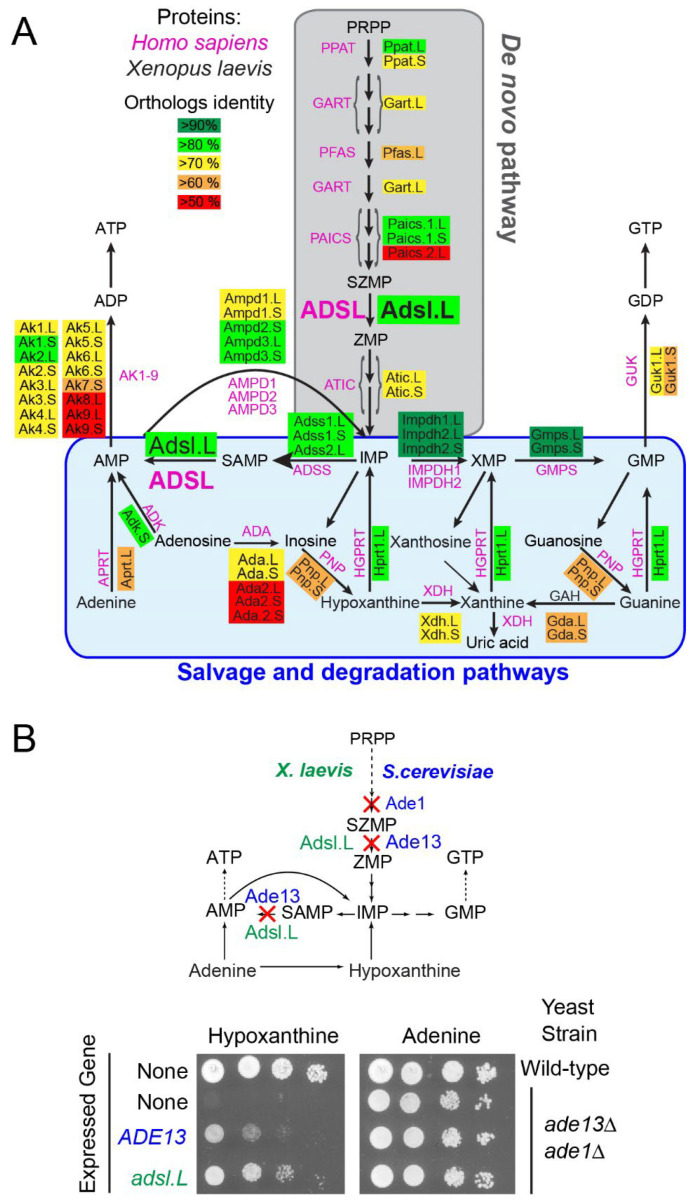
The *X. laevis adsl.L* gene encodes the adenylosuccinate lyase activity required in two non-sequential steps of the highly conserved purine synthesis pathways. (**A**) Schematic representation of the human and *X. laevis* purine biosynthesis pathways. Abbreviations: AMP, adenosine monophosphate; GMP, guanosine monophosphate; IMP, Inosine monophosphate; PRPP, Phosphorybosyl pyrophosphate; SAMP, Succinyl-AMP; SZMP, Succinyl-Amino Imidazole Carboxamide Ribonucleotide monophosphate; XMP, Xanthosine monophosphate; ZMP, Amino Imidazole CarboxAmide Ribonucleotide monophosphate. (**B**) Functional complementation of the growth defect of the yeast adenylosuccinate lyase deletion mutant via expression of the *X. laevis adsl.L* ortholog gene. Yeast wild-type and *adsl* knockout mutant (*ade13 ade1*) strains were either transformed with a plasmid, allowing for expression of the *Saccharomyces cerevisiae* (*ADE13*) or the *X. laevis* (*adsl.L*) adenylosuccinate lyase encoding genes, or with the empty vector (None). Serial dilutions (1/10) of transformants were dropped on SDcasaWA medium supplemented with either hypoxanthine or adenine as the sole external purine source. Plates were incubated for 48 h at 37 °C before imaging. Of note, the *ade1 ade13* double deletion mutant was used in this experiment to avoid genetic instability associated with the accumulation of SZMP and/or its nucleoside derivatives observed in the single *ade13* mutant [8].

**Figure 2 cells-12-02379-f002:**
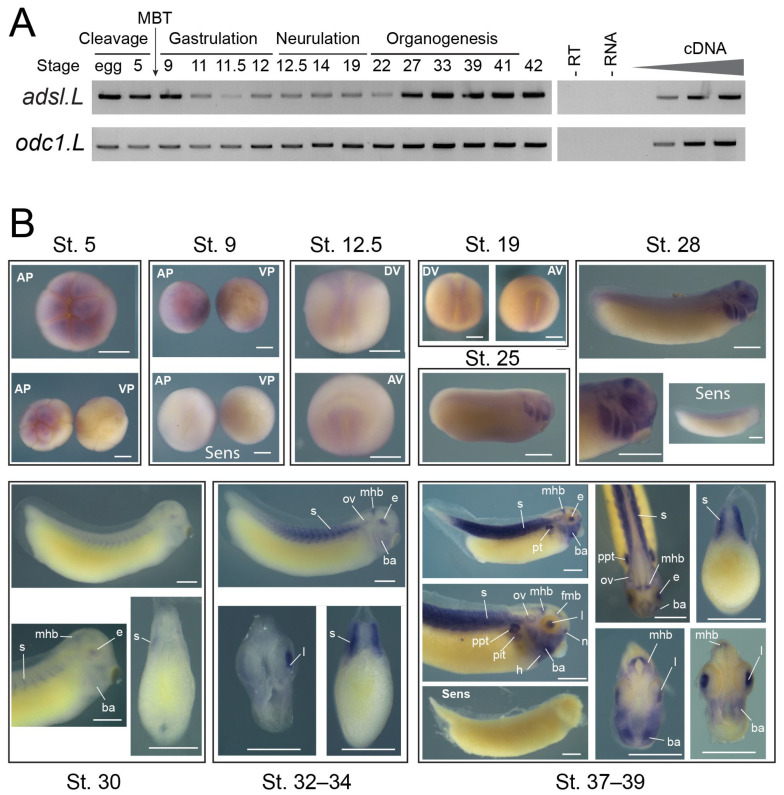
Spatiotemporal expression of *adsl.L* gene during *X. laevis* embryonic development. (**A**) Temporal expression profiles of *adsl.L* gene during embryogenesis. The expression profile was determined using RT-PCR from the cDNA of the fertilized oocyte (egg) and whole embryo at indicated stages covering the different phases of *X. laevis* embryogenesis. The ornithine decarboxylase gene *odc1.L* was used as a loading control and negative controls were performed by omitting either reverse transcriptase (-RT) or RNA (-RNA) in the reaction mix. Linearity was determined via dilutions of cDNA from stage 39 for *adsl.L* and 41 for *odc1.L*. Mid-blastula transition (MBT) is indicated. (**B**) Spatial expression profile of *adsl.L* gene during embryogenesis. Whole-mount in situ hybridization with *adsl.L*-specific DIG-labeled antisense or sense RNA probes was performed on embryos from stages (St.) 5 to 37–39. St. 5 and 9: animal (AP) and vegetal (VP) pole views, St. 12.5 and 19: dorsal (DV) and anterior (AV) views; later stages: lateral views, with dorsal up and anterior on the right, and dorsal view at stages 37–39. Transverse sections are dorsal up. Abbreviations: ba, branchial arches; e, eye; fmb, forebrain–midbrain boundary; h, heart; l, lens; mhb, midbrain–hindbrain boundary; n: nasal placode; ov, otic vesicle; ppt, pronephric proximal tubules; pit, pronephric intermediate tubules; s, somites. Bars: 0.5 mm.

**Figure 3 cells-12-02379-f003:**
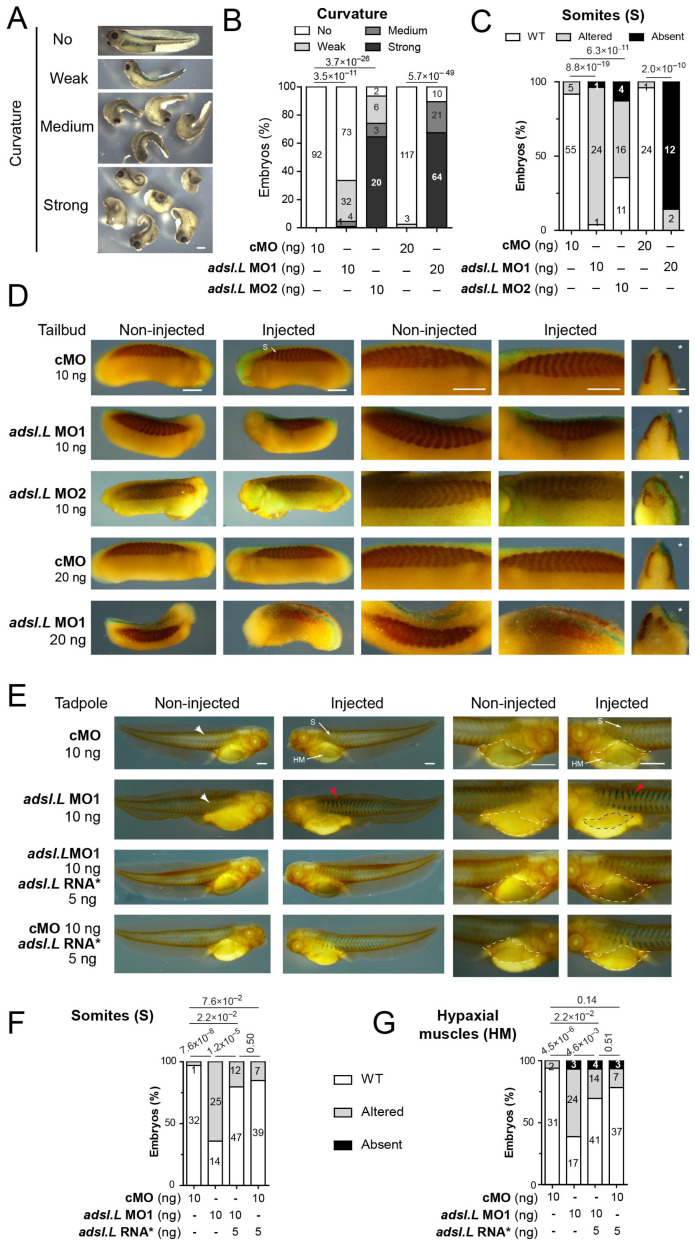
The *adsl.L* gene is required for somites and hypaxial muscle formation in *X. laevis*. (**A**) Representative images of *adsl.L* morphants following *adsl.L* MO1 injection at either 10 ng (weak and medium curvature) or 20 ng (medium and strong curvature). (**B**) Quantification and statistics of the curvature phenotype. (**C**–**G**) Immunostaining with the differentiated muscle-cell-specific 12-101 antibody revealed a strong alteration of somites and hypaxial muscle formation in *adsl.L* knockdown and *adsl.L* overexpressing embryos. Representative images (**D**,**E**) and quantification and statistics (**C**,**F**,**G**) of somites and hypaxial muscles phenotype at tailbud (**C**,**D**) and tadpole (**E**–**G**) stages. Injected side is indicated by asterisks. S, somites; HM, hypaxial muscles. Bars: 0.5 mm. White and red arrowheads point to typical somite chevron shapes and altered somites, respectively. The *adsl.L* RNA* refers to mutated RNA whose translation was not affected by the *adsl.L* specific MOs (Appendix A). Numbers above the bars of histograms correspond to *p*-values.

**Figure 4 cells-12-02379-f004:**
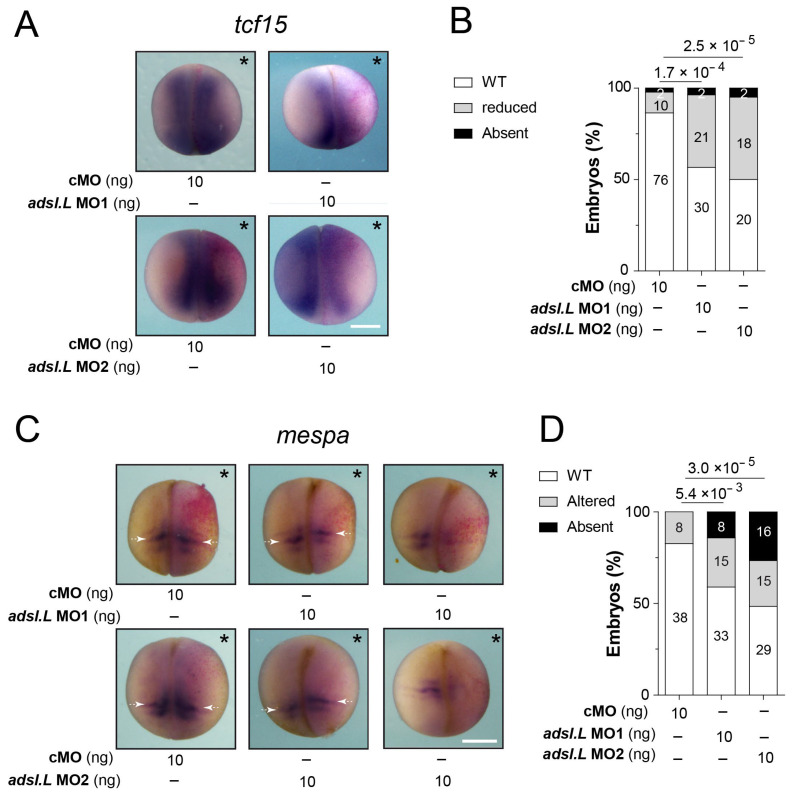
The *adsl.L* gene is required for the expression of *tcf15* and *mespa* genes involved in early somitogenesis. Expression of *tcf15* (**A**,**B**) and *mespa* (**C**,**D**) genes at late neurula stage is altered by the knockdown of *adsl.L* gene. Representative images of *trf15* (**A**) and *mespa* (**C**) RNA expression revealed via in situ hybridization. Statistics are shown in (**B**) and (**D**) for *trf15* and *mespa* RNA, respectively. White arrows point to the anterior domain of the somitomere S-II on both sides of the embryos. Injected side is indicated by asterisks. Numbers above the bars of histograms correspond to *p*-values. Bars: 0.5 mm.

**Figure 5 cells-12-02379-f005:**
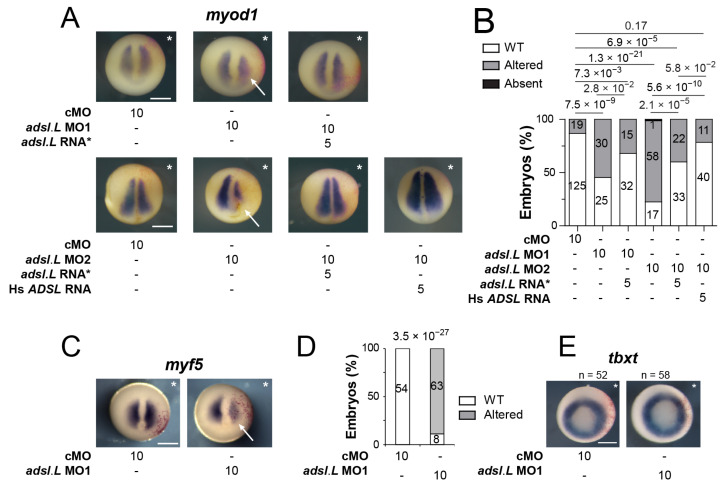
Expression of the myogenic regulatory factors *myod1* and *myf5* in paraxial mesoderm was strongly affected by the knockdown of the *adsl.L* gene. (**A**) Expression of *myod1* gene at the early neurula stage was altered by the knockdown of *adsl.L* gene and rescued by *adsl.L* RNA* and human *ADSL* RNA. (**C**) Representative images of the effect on *myf5* RNA expression by *adsl.L* knockdown at stage 12.5 revealed via in situ hybridization. (**B**,**D**) Quantification and statistics of the *myod1* and *myf5* expression phenotypes are presented in (**A**,**C**), respectively. (**E**) Knockdown of *adsl.L* did not alter the mesoderm formation, as revealed by the absence of change in *tbxt* (*xbra*) RNA expression domain at stage 11. Injected side is indicated by asterisks. The amounts of MO and RNA presented in this figure are in ng. Bars: 0.5 mm. Numbers above the bars of histograms correspond to *p*-values. White arrows point to the domain where expression of either *myod1* or *myf5* is altered.

**Figure 6 cells-12-02379-f006:**
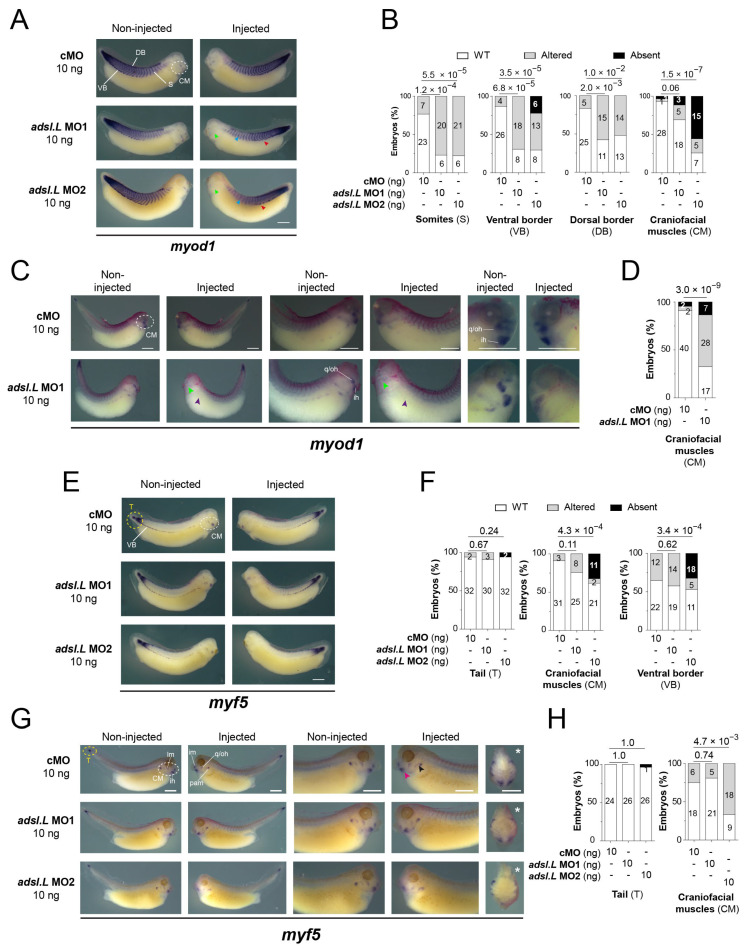
Knockdown of *adsl.L* gene caused an impaired expression of *myod1* and *myf5* at late tailbud and tadpole embryonic stages, leading to somites and craniofacial muscle formation defects. (**A**–**D**) Representative images of the *adsl.L*-dependent altered expression of *myod1* (**A**) and *myf5* (**C**) domains in either somite (S), dorsal somite border (DB), ventral somite border (VB) or craniofacial muscles (CM), as revealed via in situ hybridization in late tailbud embryos. Blue, red, green and purple arrowheads point to *myod1* altered expression in somites, ventral border, craniofacial muscles and hypaxial muscles, respectively. (**E**,**G**) Representative images of the *adsl.L*-dependent altered expression of *myod1* (**E**) and *myf5* (**G**) domains in the unsegmented mesoderm in the most posterior tail (T), somites (S), ventral somite border (VB) and craniofacial muscles (CM) in tadpole embryos. Black and pink arrowheads point to *myf5* altered expression in pharyngeal arch muscle enlagen intermandibularis muscle enlagen respectively. (**B**,**D**,**F**,**H**) Quantification and statistics of the *myod1* and *myf5* expression phenotypes presented in (**A**,**C**,**E**,**G**), respectively. Craniofacial muscles: ih, interhyoïedus anlage; im, intermandibularis anlage; lm, levatores mandibulae anlage; pam, pharyngeal archmuscle anlagen; q/oh, common orbitohyoïdeus and quadrato-hyoangularis precursors. Injected side is indicated by asterisks. Bars: 0.5 mm. Numbers above the bars of histograms correspond to *p*-values.

**Figure 7 cells-12-02379-f007:**
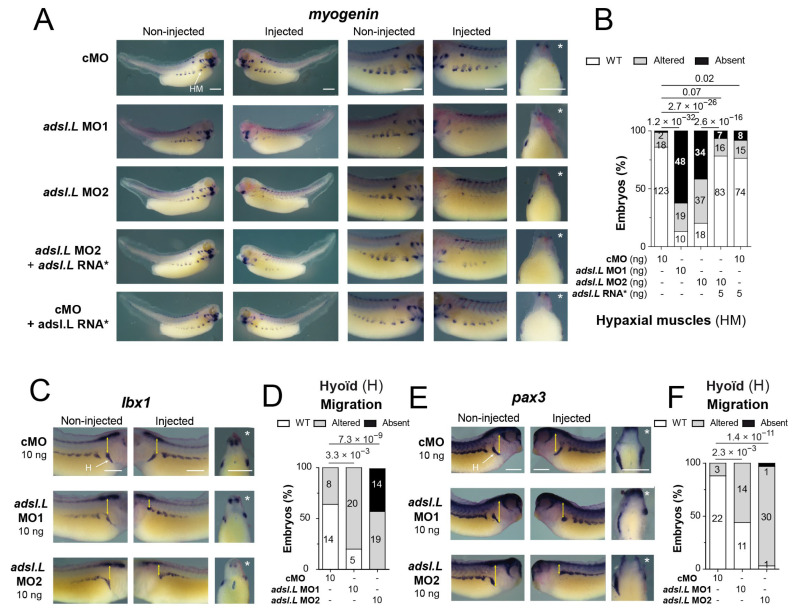
The *adsl.L* gene is required for hypaxial muscle migration. (**A**,**B**) The hypaxial muscle defect associated with *adsl.L* knockdown was rescued by the MO non-targeted *adsl.L* RNA*. Representative images (**A**) and quantifications (number in bars) and statistics (**B**) of the *adsl.L*-dependent alteration of *myogenin* gene expression monitored via in situ hybridization. (**C**–**F**) Migration of myoblasts in the hyoid region was found to be severely affected in *adsl.L* morphants, as shown by *lbx1* (**C**,**D**) and *pax3* (**E**,**F**) gene expression pattern alterations. Injected side is indicated by asterisks. Bars: 0.5 mm. HM and H stand for hypaxial muscles and hyoid region, respectively. Numbers above the bars of histograms correspond to *p*-values.

## Data Availability

Data sharing is not applicable to this article.

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
