# Peer review of "Purine Biosynthesis Pathways Are Required for Myogenesis in Xenopus laevis"

_cells, 2023, doi:10.3390/cells12192379_

Round 1

Reviewer 1 Report (Previous Reviewer 1)

In the discussion, you can specify the similarities with the RARβ2 morphants: a rostral shift of presomitic markers, defects in somite morphology, chevron-shaped somites, and hypaxial muscle formation. Similarly, you can also mention that hypaxial muscle defects have already been observed in knockout mice for paraxis.

Wilson-Rawls J, Hurt CR, Parsons SM, Rawls A. Differential regulation of epaxial and hypaxial muscle development by paraxis. Development. 1999 Dec;126(23):5217-29. doi: 10.1242/dev.126.23.5217. PMID: 10556048.

Author Response

As requested, comments and reference have been added, see ref n°50 and text modification page 23 and text modifications page 27.

Reviewer 2 Report (Previous Reviewer 2)

The manuscript has been greatly improved and my concerns were addressed by the authors.

The only suggestion I have is to avoid using the word "significantly" when referring differences by eye (comparison of in situ hybridization patterns) as for example: "Expression of myf5 was also significantly altered in craniofacial muscles at both the tailbud and tadpole stages...".

It is much better to use words such as "considerably", "substantially", "markedly" or "noticeably".

The English has been greatly improved.

Author Response

The “significantly” term had been used to refer to statistical analyses, but has been changed by synonyms according to your suggestion (see text page 14 line 5, page 19 penultimate line, and page 20 lines 3, 10 and 12).

This manuscript is a resubmission of an earlier submission. The following is a list of the peer review reports and author responses from that submission.

Round 1

Reviewer 1 Report

Muscular and neurological disorders are major symptoms of genetic diseases associated with purine pathway genes. This is the first report describing the involvement of purine pathway genes in the control of myogenesis during development.

Enzymes of the purine biosynthesis pathway, particularly ADSL, were studied during myogenesis of the species Xenopus laevis. First, the functional characterization of these genes was carried out by complementation experiments in yeast. Next, expression of purine pathway genes in Xenopus embryos was described by RT-PCR and whole-mount in situ hybridization (WISH), predominantly in somites and the nervous system. The involvement of ADSL during myogenesis was shown by loss-of-function experiments using two translation-blocking morpholinos. Whole-mount immunochemistry with the 12/101 antibody was used to highlight the impairment of somitogenesis. Expression defects of the myogenic factors myf5, myod1, and myogenin (MRF) were also observed during primitive, hypaxial, and somite-derived craniofacial myogenesis using WISH. Precocious markers of hypaxial myogenesis pax3 and lbx1 revealed formation or migration deficiencies of myogenic progenitors. Similar defects were also described with translation-blocking morpholinos of two enzymes of the salvage and de novo purine pathways, respectively, hprt1.L and ppat. Rescue experiments confirmed the specificity of the morpholinos. Moreover, gain-of-function experiments using human or Xenopus synthetic mRNAs activated myod1 expression at the neurula stage. The authors suggest that purine pathway genes may contribute to the activation of signaling pathways involved in the control of MRF expression during somitic and hypaxial myogenesis.

General comments:

The obvious hypaxial defect was demonstrated by loss-of-function experiments against three genes of the purine pathways using translation-blocking morpholinos. Rescue experiments with the corresponding mRNAs confirmed the specificity of the morpholinos. However, no other loss-of-function methods were used to confirm the phenotype. It seems that the dominant negative form of ADSL has not been characterized, but could an inhibitor of the purine pathway be tested? Concerning the morpholinos experiment, the control and rescue experiments were generally well done. However, the control morpholino was not an inverted or mismatch morpholino but a standard control. Moreover, the two morpholinos used for ADSL were partly overlapping. The rescue experiment was only carried out with a high mRNA dose (5 ng), and no other methods were carried out to confirm the rescue. As pointed out in the discussion, rescue by adenine treatment has failed, but have you tried injecting adenine into the blastocoel or performing the treatment on dechorionated embryos? So although it is not necessary to carry out all these control experiments, it is better to carry out at least one of them.

Morpholinos against purine pathway genes also alter primitive myogenesis and somitogenesis. It cannot be excluded that a defect in the formation or migration of muscle progenitors at the hypaxial level could be a consequence of the alteration of primitive myogenesis and somitogenesis. The morphants of the purine biosynthetic pathway exhibit similarities to those of RARbeta2 (Janesick et al., 2017), showing defects in somite morphology, chevron-shaped somites, and hypaxial muscle formation. Moreover, there may also be differences in somite size and number (Fig 3D, MO1 and 2, Fig 5A, MO2). To better characterize the phenotype of the morphants, the expression of genes involved in somitogenesis should also be analyzed and discussed. Alternatively, a rescue experiment could be conducted with Myod1 synthetic mRNA and analyzed with the 12/101 antibody to determine whether decreased Myod1 expression is the main cause of the phenotype, as suggested in the discussion. A better characterization of the phenotype could highlight the results of this promising study.

Specific comments:

P6, 2.5. mRNA synthesis and morpholino oligonucleotides: “The position of the MOs in relation to their respective cDNAs is indicated in the Figure S3”. It is RNA not DNA.

P12 fig3: To facilitate the reading of the manuscript, please specify in the legend of the first figure or on the figure itself that the numbers above the bars of the histogram correspond to the p-value.

P15: “The alteration of these muscles is consistent with the fact that these craniofacial muscles de-rived from the anterior somites, highly altered in adsl.L knock-down conditions (Figure 3 and Figure Insert reference.

P16 Fig5C: Images that depict abnormalities of head myogenesis in Fig. 5C are not always clear. A front view or higher magnification could be used to better illustrate the phenotype.

P20: “Muscle defects associated with purine pathways dysfunction are related to an altered expression of MRF genes”. Insert reference

P21: “We can therefore speculate that the involvement of de novo and salvage purine pathways during this myogenic wave is conserved in mammals and that purines may therefore be key regulators in the formation of hypaxial myogenic cells, the progenitors of abdominal, spinal, and limb muscles, those muscles that are affected in patients with purine-associated pathologies”. Insert reference.

Reviewer 2 Report

Purine-associated pathologies have a wide spectrum of clinical symptoms, but severe muscular and neurological dysfunctions are common in patients with mutations in genes involved in purine biosynthesis. The manuscript by Duparray et al. addresses the usefulness of using the Xenopus laevis model to (1) identify the expression pattern of genes involved in purine biosynthesis during embryonic development and (2) to determine the effect of morpholino knock-down of selected genes on the development of specific tissues. They focussed mostly on the adsl.L gene in Xenopus as a proof of concept. This is a good choice since ADSL is involved in both the de novo and salvage pathway.

The authors demonstrate that these Xenopus genes are able to rescue yeast with mutations in the orthologous genes, they determine the temporal and spatial gene expression profile of these genes and then injected morpholinos targeting adsl.L in one of the blastomeres of 2 cell stage Xenopus embryos, thus perturbing the expression of adsl.L on one of the sides of the embryos. They then addressed the effect of morpholino injection on muscle formation and concluded that myotome, hypaxial and craniofacial muscle development was perturbed in the morphants. They also used appropriate controls in these experiments. They further go on to characterize the expression of the MRFs as well as genes expressed in migrating muscle cells, to pinpoint when muscle development starts being different on the experimental side.

This is an interesting study that tries to contribute knowledge towards understanding the developmental origin of purine-associated pathologies, focussing on skeletal muscle. Some might argue that a better approach would be to use human cells or mouse. However, the purine biosynthesis pathways are highly conserved. So, the more models are studied, the better, for understanding how perturbing these pathways leads to disease. The Xenopus system has the advantage of being a solid developmental model, which is relatively easy to manipulate, so a considerable volume of data was accumulated in this paper which gives a first step towards understanding the role of purine biosynthesis in myogenesis.  More work has to be done, of course, to understand these pathologies, but this paper is an important contribution, and the methodology can be replicated in the future addressing other genes and/or tissues.

There are, however, certain aspects of the paper that should be improved and I list those below:

1. The RT-PCR experiments determining the temporal expression of adsl.L (Figure 2A) should not be interpreted in a quantitative way. I suggest modifying the sentence “Adsl.L is expressed in all developmental stages tested, with a high level of maternal expression (before MBT). Its zygotic expression level increases from early neurula stages and again during organogenesis, to reach the peak of expression during later organogenesis.” (p. 8) to remove words such as high level, increases and peak. One can conclude that there is maternal expression (before MBT) and also embryonic expression in all stages studied.

2.     All the in situ hybridization experiments in the paper are very well done and the images are clear, although in Figures 3 and 4 they are a little bit small. In Figure 2B they are a good size. However, I suggest some care in using quantitative comparisons between different embryos or experiments. Moreover, the text referring to Figure 2B is not very clear. Particularly the phrase “From stage 30 onwards, Adsl.L transcripts are also detected in mesoderm derivative tissues (should be mesoderm-derived tissues), the pronephric tubules (in the pronephric proximal and intermediate tubules), the heart and mostly in the somites and its somitic expression level increases during organogenesis.” I don’t quite understand the use of the word “mostly”. Do you mean “From stage 30 onwards, Adsl.L is mostly expressed in the somites”, in that that is where most of the in situ signal is? Then perhaps the best is to turn the sentence around and start with the somites, and then mention the other mesoderm-derived tissues later in the sentence. Also, I suggest either removing the part about expression level increasing during organogenesis, since many variables can contribute to a stronger signal, or introduce some doubt, e.g. “appears to increase”.

3.     The process of somitogenesis sometimes appears as synonym to myogenesis, which is a little confusing for the reader. The title of the paper uses the term myogenesis, but in other sections, somitogenesis is sometimes used. In anamniotes, myotome differentiation occurs before or simultaneously with somitogenesis, but they are still distinct processes (somitogenesis is border formation and acquisition of the somite shape and myotome formation involves myocyte differentiation). So I suggest avoiding mixing the two words as if they are synonyms. Examples of sections where that can be improved are: (1) on p. 10 where the authors say “This curvature phenotype was already documented when somitogenesis was altered, suggesting that adsl.L gene could be required for proper myogenesis in Xenopus.” For example, say “could be required for proper somitogenesis and myogenesis in Xenopus”. Or use only the word “myogenesis”. (2) Another case is the figure legend of Figure 3. Here the authors say that the adsl.L gene is required for somitogenesis and hypaxial muscle formation. With somitogenesis the authors mean formation of the chevron-shaped form with clear boundaries? How about myogenesis in the somite, is it also affected? (3) This confusing terminology is again present in the Discussion (p. 20; section that starts with “In addition, …” as well as in the section that starts with “Our study provides….”. Perhaps introducing a clear definition of these terms in the M&M or in this results section would help to clarify what is somitogenesis and what is somitic myogenesis.

4.     In section 3.3. of the results, the phenotype of the MO-injected side is compared to the control side after staining with antibody recognizing a muscle antigen. However, this section would benefit from more precise language. For example, in the phrase “myofibers were often reduced along the dorsoventral axis” it is not clear what myofibers being reduced means. Perhaps you mean that myotomes (and not the myofibers) are smaller along the dorsoventral axis? Also, the authors mention that “the myotome area is thinner”, is this being thinner in the medio-lateral direction or more space between cells? I suggest using arrows pointing to e.g. dorsal-most positive signal and ventral-most positive signal on each image or use ] or } symbols on the images so that the reader can visually see in which direction the reduction in size occurs, as well as using more precise language. Another example is on p 14, when the authors say “… myod1 was found significantly reduced in adsl.L MO1 and MO2 morphants, with myogenic fibers shortened along the dorsoventral axis and with strong alterations and even loss of the chevron shape….”. If I understand correctly, the myofibers at this stage are oriented in an anteroposterior direction, parallel to the notchord/neural tube. So it doesn’t make sense saying the fibers are shortened along the dorsoventral axis. Maybe the authors mean that the myotomes are shortened along the dorsoventral axis?

English corrections (note that I am not a native English speaker, but here are some suggestions but review by a native speaker or grammar programme would be best):

1.     The authors use various ways of writing Xenopus laevis: Xenopus laevis, X. laevis, Xenopus or xenopus. Please do a global replace to remove xenopus. The other three are correct, but maybe not use all three.

2.     The manuscript would benefit from an English grammar revision. The English is good, but the grammar is sometimes not correct which is a shame. I list some examples below, but there are more:

3.     P.1 Abstract, replace: “…leads to severe reduction of the Myogenic Regulatory Factor’s expression…” with “…leads to severe reduction in the expression of Myogenic Regulatory Factors …”

4.     P.1, replace “…such as acids and lipids biosynthesis…” with “…such as the biosynthesis of acids and lipids …”

5.     P.4 Figure legend, replace “…highly conserved purines synthesis pathways…” with “…highly conserved purine synthesis pathways…”

6.     P. 4, replace “…blood, urines, and…” with “…blood, urine, and…”

7.     P. 5, replace “…results in down-expression at different developmental stages of several myogenic regulating factors (MRFs)…” with “…results in down-regulation (or reduction in expression) of several myogenic regulatory factors (MRFs) at different developmental stages …”

8.     P. 5, replace “…in craniofacial and hypaxial muscles formation…” with “…in craniofacial and hypaxial muscle formation…”

9.     P.5. Title 2.4 “Embryos Culture” should be “Embryo Culture” and remove italic from the text in that section.

10.  P.6., replace “…in relation to their respective cDNAs is indicated in the Figure S3..” with “…“…in relation to their respective cDNA is indicated in Figure S3..”

11.  P. 10, replace “…a similar expression temporal profile.” with “…“…a similar temporal expression profile.”

12.  P. 12. Figure 3A; to describe the curvature, I suggest using the words Weak, Medium, Strong.

13.  P.21, replace “…observed in the C2C12 mice cells” with “…observed in the mouse C2C12 cell line”

14.  P.21 The phrase “To our knowledge, a similar direct regulation of MRFs by purines, that could suggested by our data, is not described to date, and if it exists, it occurs via a mechanism that remains to be established” is long and unclear.  A suggestion is: “A similar direct regulation of MRFs by purines has not been described to date. If it exists, which our data suggest, the mechanism by which it acts remains to be established.”

Round 2

Reviewer 1 Report

The results of the paper "Purine biosynthesis pathways are required for myogenesis in Xenopus laevis" are certainly new and promising. However, the description of the phenotype of morphants demonstrating the involvement of purine biosynthesis pathways in the development of hypaxial muscles during somitogenesis remains incomplete, given the current knowledge on the subject. As a review including new experiments is not possible in this journal, I recommend a resubmission that includes an analysis of the expression of genes involved in somitogenesis